

# Influence of tied-ridge with biochar amendment on runoff, sediment losses, and alfalfa yield in northwestern China

Erastus Mak-Mensah[1], Faisal Eudes Sam[2], Itoba Ongagna Ipaka Safnat Kaito[3], Wucheng Zhao[1], Dengkui Zhang[1], Xujiao Zhou[1], Xiaoyun Wang[1], Xiaole Zhao[1] and Qi Wang[1]

[1] College of Grassland Science, Gansu Agricultural University, Lanzhou, Gansu Province, China
[2] College of Food Science and Engineering, Gansu Agricultural University, Gansu Key Laboratory of Viticulture and Enology, Lanzhou, Gansu Province, China
[3] College of Science, Gansu Agricultural University, Lanzhou, Gansu Province, China

Corresponding author
Qi Wang, wangqigsau@gmail.com

## ABSTRACT

**Background**. Loss of organic matter and mineral nutrients to soil erosion in rain-fed agriculture is a serious problem globally, especially in China's Loess Plateau. As a result, increasing rainwater usage efficiency by tied-ridge-furrow rainwater harvesting with biochar is expected to improve agricultural productivity. Nonetheless, with limited knowledge on tied-ridge-furrow rainwater harvesting with biochar, small-scale farmers face the challenge of adoption, thus, the rationale for this study.

**Materials and methods**. A field experiment was conducted to determine the influence of open-ridging (OR) and tied-ridging (TR) with bio-degradable film on ridges and biochar in furrows on runoff, sediment losses, soil moisture, fodder yield, and water use efficiency (WUE) on sloped land, using flat planting (FP) without ridges and furrows as control, during alfalfa-growing year (2020).

**Results**. Runoff in flat planting (30%), open ridging (45%), and tied ridging (52%) were decreased with biochar to the extent where sediment was decreased in flat planting (33%), open ridging (43%), and tied ridging (44%) as well. The mean runoff efficiency was lower in flat planting (31%), open ridging (45%), and tied ridging (50%) in biochar plots compared to no-biochar plots. In biochar and no-biochar plots, soil temperature on ridges of TR was higher than that on OR, which was higher than FP during alfalfa growing season. Soil temperature in furrows during alfalfa growing season in biochar and no-biochar plots were in the order FP > OR > TR. Mean soil water storage for FP, OR, and TR, in biochar plots was higher than in no-biochar plots. This indicates biochar has a beneficial impact on open riding. Total annual net fodder yield (NFY) was significantly ($p = 0.00$) higher in treatments in the order TR > OR > FP. Tied ridging had a significant effect on actual fodder yield (AFY) in biochar plots, while open ridging significantly affected AFY in no-biochar plots. Annual total mean NFY and AFY increased by 8% and 11% in biochar plots compared to no-biochar plots. In biochar and no-biochar plots, water use efficiency was in the order TR > OR > FP. Conclusively, water use efficiency was significantly higher ($p = 0.01$) in biochar plots compared to no-biochar plots.

**Conclusion**. When crop production is threatened by soil erosion and drought, mulched tied-ridge with biochar is beneficial to crop growth in rain-fed agriculture, according to this research. Smallholder farmers should be trained on applying this technique for

water-saving to mitigate runoff, soil erosion, sediment losses, and improve food security in semiarid areas.

## INTRODUCTION

The Loess Plateau of China is accustomed to unpredicted rainfall, with spatial variations accompanied by recurring droughts (*Ding et al., 2018*; *Wang, Jia & Liang, 2015*; *Jin et al., 2007*). In addition to these climatic conditions of the Loess Plateau, topography contributes immensely to severe soil erosion in events of heavy rainfall (*Chen et al., 2019*; *Wang, Xiong & Kuzyakov, 2016*; *Meng et al., 2008*). Consistent soil erosion reduces soil nutrient and moisture availability for crops (*Jin et al., 2007*; *Fu et al., 2004*; *Trimble & Crosson, 2000*). Thus, loss of soil nutrients destabilizes the structure and population of microbes in the soil (*Xiao et al., 2017*; *Du et al., 2020*). Consequently, alfalfa (*Medicago sativa L.*) was revealed as ideal for protecting soil hence, was commonly grown on the Loess Plateau (*Jun et al., 2010*). Soon after, alfalfa was discovered to be seriously depleting soil water in these areas, due to its deep roots' water absorption from deep soil layers (*Jun et al., 2010*). *Fan et al. (2016)* reported alfalfa can exhaust available soil water in a field in less than 6 years and prevent deep soil water recharge. *Wang et al. (2012)* corroborated this by revealing dry soil layers in deep soil after alfalfa cultivation which has significantly obstructed sustainable agriculture development. However, these fragile regions remain the main source of livelihood for millions of deprived and vulnerable people (*Bado, Whitbread & Sanoussi Manzo, 2021*). On this account, agriculture in semiarid areas must be improved by increasing rainwater use efficiency with mulching in rainwater harvesting scheme (*Meng et al., 2020*).

In recent years, many field studies have revealed ridge and furrow rainwater harvesting (RFRH) as an effective and simple technique for increasing soil water content (SWC) and improving rainwater use efficiency in rain-fed agriculture (*Li et al., 2016*; *Xiaolong et al., 2008*). RFRH can gather effective or ineffective rainfall, prevent surface runoff during intensive rainfall, and reduce evaporation (*Zheng et al., 2019*; *Jia et al., 2018*; *Liang et al., 2018*; *Han et al., 2013*). RFRH is extensively practiced in areas with <5 mm rainfall, where irrigation is not available (*Liu et al., 2020*) for improving rainfall infiltration and soil moisture (*Ren et al., 2016*), facilitating seedling growth at a faster pace (*Gan et al., 2013*; *Zhang et al., 2011*; *Ramakrishna et al., 2006*), and improving crop yield and maintaining food stability (*Chen et al., 2015*; *Bu et al., 2013*). RFRH has been demonstrated in many studies to increase soil temperature as compared to flat planting (*Zhang et al., 2017*; *Mo et al., 2017*). In extreme rainfall events, however, water runs over the ridges (*Wiyo, Kasomekera & Feyen, 2000*). Hence, building basins with cross-ties known as tied-ridging, to store surface runoff in furrows is a solution to excess water flowing over ridges (overtopping) in RFRH on sloped lands (*Vejchar et al., 2019*). The collected water can be used by crops for a long time better than it can be used in the state of runoff (*Ndlangamandla, Ndlela*

& Manyatsi, 2016). The cross-ties also reduce the speed of the water flow along furrows (Mutiso, 2018) and often increase the length of crop growing seasons (Mason et al., 2015).

Previous studies have shown that tied-ridge, which is a proven method of maintaining soil moisture at 0–5 and 6–10 cm soil depth in drier periods in rain-fed agriculture, increases yield by 50% (Mandumbu et al., 2020; Sibhatu et al., 2017). Beshir & Abdulkerim (2017) revealed an increase in soil fertility with in-furrow planting in a closed-end tied-ridge system. Consequently, Mupangwa, Love & Twomlow (2006) reported an average maize yield of 3,400 kg ha⁻¹ from tied ridges compared with 1,500 kg ha⁻¹ from conventionally ploughed fields. A study conducted by Brhane et al. (2006) revealed that variations in tied-ridging beneficial effects on crop yield are due to differences in distribution and amount of rainfall, slope, soil type, landscape position, time of ridging, and crop type. They further stated that soil water and sorghum grain yield was increased with tied-ridging by more than 25 and 40%, respectively, as compared to conventional tillage (shilshalo) practice in northern Ethiopia. Tied-ridging has been effective in increasing soil water storage and decreasing runoff in Tanzania (Guzha, 2004), and the USA (Howell, Schneider & Dusek, 2000). However, inappropriate use of tied-ridging can lead to problems such as waterlogging, and total loss of crops in harsh storms (Brhane et al., 2006). Studies in arid and semiarid areas of sub-Saharan Africa suggested that single water conservation interventions could improve crop yields by up to 50% (Araya & Stroosnijder, 2010; Bennie & Hensley, 2001; Walker, Tsubo & Hensley, 2005) while combination of tied-ridges and nutrient inputs have accounted for two-fold to six-fold crop yields compared with conventional tillage practices without fertilizer use (Jensen et al., 2003; Zougmoré, Zida & Kambou, 2003). Therefore, given the deficient soil fertility nature of arid and semiarid areas of northwestern China, single rainwater harvesting intervention may not bring about a considerable influence on crop productivity (Biazin & Stroosnijder, 2012). Thus, tied ridging with mulching which has been widely practiced in many countries (Donjadee & Tingsanchali, 2016; Chakraborty et al., 2008; Mupangwa, Love & Twomlow, 2006) with a consistent increase in crop production should be explored. Mulching, a significant agronomic practice, is gaining considerable attention globally due to its phenomenal effect and low cost (Li, Li & Pan, 2020). Mulching has different generally established environmental functions (Prosdocimi, Tarolli & Cerdà, 2016). Some of which are notably increasing soil surface coarseness hence decreasing runoff, sediment, and nutrient content in runoff (Vega, Fernández & Fonturbel, 2015; Lee et al., 2018). In addition, mulching retains soil moisture, hence increases rainfall infiltration and decreases evapotranspiration (Li, Li & Pan, 2020). Decomposed mulching materials increase soil organic matter and available soil nutrients for crop development (Jiménez et al., 2016; Bajgai et al., 2014; Jordán, Zavala & Gil, 2010). There have been significant reports on effectiveness of mulching in reducing soil water and nutrient loss in different climatic environments in America (Ruy, Findeling & Chadoeuf, 2006), Europe (Fernández et al., 2012; Abrantes et al., 2018), Asia (Wang, Xiong & Kuzyakov, 2016), and Africa (Mwango et al., 2016). One such prominent mulching technology is biochar amendment (Woolf et al., 2010; Woolf et al., 2018).

Biochar, a steady carbon-rich material manufactured from pyrolyzing biomass in oxygen-deprived environments, can improve soil carbon sequestration and soil quality

(*Lehmann & Rondon, 2006*). A potential feedstock is shelled maize cobs crop residues, often burnt or left on the field in rural regions of developing countries to decompose (*Silayo et al., 2016*). By improving cation exchange capacity and soil structure, biochar increases soil fertility (*Martinsen et al., 2014*) and decreases nutrient leaching (*Laird et al., 2010*). Biochar became known as a key element of the popular fertile anthropogenic Terra Preta soil of Central Amazonia (*Glaser & Birk, 2012*). Studies have confirmed biochar as extremely viable for curbing soil and nutrients losses on sloping lands in semiarid regions (*Li et al., 2019*; *Zhang et al., 2017*; *Han, Ren & Zhang, 2016*; *Xiao et al., 2016*; *Liu, Han & Zhang, 2012*). For example, *Kammann et al. (2012)* discovered a significant increase in biomass in biochar-modified soils relative to controls in perennial ryegrass (*Lolium perenne L.*). Consequently, *Rondon et al. (2007)* revealed addition of biochar to a low-fertility soil led to 22% increased nitrogen fixation in beans (*Phaseolus vulgaris*) in addition to significantly improved biomass and bean yield. In terms of runoff and erosion, biochar can help increase infiltration rate and saturated hydraulic conductivity (Ksat) in clayey soils thereby curbing erosion, flooding, and contamination of streams (*Li et al., 2019*; *Li et al., 2018*; *Obia et al., 2018*; *Lim et al., 2016*). Saturated hydraulic conductivity is the ease of water flow through the soil when it is saturated and it is vital for flooding, drainage, and soil water studies (*Lu, 2015*; *Kirkham, 2014*). Biochar has also been recounted to improve soil physical and hydrological properties, ranging from bulk density and soil porosity to soil aggregate stability (*Fischer et al., 2019*; *Burrell et al., 2016*; *Glab et al., 2016*). Biochar amendment, in combination with a slow decomposition (*Peng et al., 2011*; *Wang, Xiong & Kuzyakov, 2016*), foster carbon sequestration and long-term soil improvements (*Kuzyakov, Bogomolova & Glaser, 2014*; *Lehmann et al., 2008*), and thus can aid in mitigating climate change (*Crane-Droesch et al., 2013*; *Woolf et al., 2010*). *Jeffery et al. (2017)* discovered in a meta-analysis that extremely predominantly weathered soils, prevalent in the humid tropics, benefit from biochar amendments with mean crop yield increases of 25%. Meanwhile, in some other studies, biochar has been demonstrated not to influence soil moisture. *Hardie et al. (2014)* recounted that 30 months after biochar amendments to a sandy loam soil, no significant outcome was revealed on soil moisture at various tensions (measurement of the quantity of energy necessary to transport water in the soil). Conversely, *Gonzaga et al. (2018)* found that soils treated with 30 t ha$^{-1}$ coconut husk biochar increased 90% of Zea mays biomass, while orange bagasse biochar applied at the same concentration had no impact. The disparity in outcomes from different studies, however, could be ascribed to differences in soil types, plant species treated, biochar application rates, and experimental circumstances (*Edeh, Mašek & Buss, 2020*; *Nooker, 2014*).

In a recent study, *Anyanwu et al. (2018)* found aged biochar in soil has a detrimental impact on earthworms and/or fungi growth. Furthermore, this resulted in a decrease in rice (*Oryza sativa*) and Tomato (*Solanum lycopersicum*) underground root biomass. In addition, biochar has been shown to reduce soil thermal diffusivity due to biochar's low thermal diffusivity (*Zhao et al., 2016*). Biochar's beneficial effects are shown to be soil specific, contrary to common belief. As a result, biochar amendment could not be beneficial to all forms of soil (*Zhu, Peng & Huang, 2015*). Nevertheless, when biochars were used, several studies identified weed problems. Biochar application at relatively high rates of 15 t

ha$^{-1}$ resulted in a 200% increase in weed growth during lentil culture, according to *Safaei Khorram et al. (2018)*; suggesting repeated biochar applications might not be good for weed control. According to *Vaccari et al. (2015)*, applying 14 t ha$^{-1}$ of biochar to tomato plants improved vegetative growth but not fruit yield. Instead of providing plant nutrients, biochar can react with soil nutrients and function as a competitor (*Joseph et al., 2018*). Biochar can adsorb nitrogen as well as essential nutrients like Fe, which can be detrimental to plant development (*Kim et al., 2015*), since this may delay plant flowering (*Hol et al., 2017*). Biochar amendments in saline sodic soil could aid phosphate precipitation and sorption reactions which could ultimately lead to a reduction in amount of phosphorus available to plants (*Xu et al., 2016*). Concurrently, biochar amendment in soil, for example, had no effect on pesticide absorption of dichlorodiphenyltrichloroethane (DDT) (*Denyes, Rutter & Zeeb, 2016*). In terms of soil biology, biochar can disrupt organic matter decomposition, reducing abundance of fungi species such as Ascomycota and Basidiomycota by 11 and 66%, respectively (*Zheng et al., 2016*). Despite several studies showing biochar amendment has positive and negative effects, there is still a lot of confusion regarding effects in conjunction with other management techniques (*Solaiman & Anawar, 2015*).

Although biochar amendment (*Solaiman & Anawar, 2015*), and tied-ridging (*Twomlow & Bruneau, 2000*) has widely been explored, conflicting reports on the effects in conjunction with other management techniques are prevalent. In addition, most studies report on only yield advantages, ignoring trade-offs between runoff, sediment losses, soil temperature, and moisture (*Ademe, Bekele & Gebremichael, 2018*). Therefore, there is the need for investigations into the combined effects of tied-ridge as a field soil moisture conservation technique with biochar on sloped land is needed to allow this know-how to be better situated to compete with other droughts, and soil erosion mitigation approaches (*Woolf, Lehmann & Lee, 2016*; *Woolf et al., 2018*). To date, however, worldwide experiments are relatively rare to enumerate capabilities of tied-ridge with biochar on sloped lands, in terms of their capacity to guarantee food security and dealing with extreme conditions, such as drought. This study reports the influence of tied-ridge with biochar amendment on soil temperature, moisture, runoff, sediment losses, and alfalfa fodder yield. The specific objectives of this study were (1) to determine whether biochar amendment in tied ridging reduces soil temperature, runoff, and sediment losses on sloping lands, and (2) to determine optimum mulch recommendation with tied ridging that will produce high alfalfa yield and water use efficiency in semiarid Loess Plateau of China.

## MATERIALS AND METHODS
### Schematic overview of the experimental program
The schematic overview of the experimental program, from identification of the experimental station, to sampling and measurements of alfalfa cultivation in tied-ridge with biochar amendments, is displayed in Fig. 1. For emphasis, the purpose of this research was to examine the influence of tied-ridge with biochar amendment on runoff, sediment losses, and alfalfa yield in northwestern China. The study implemented a completely randomized design with three replications. Tied-ridging, open-ridging, and flat planting were the three

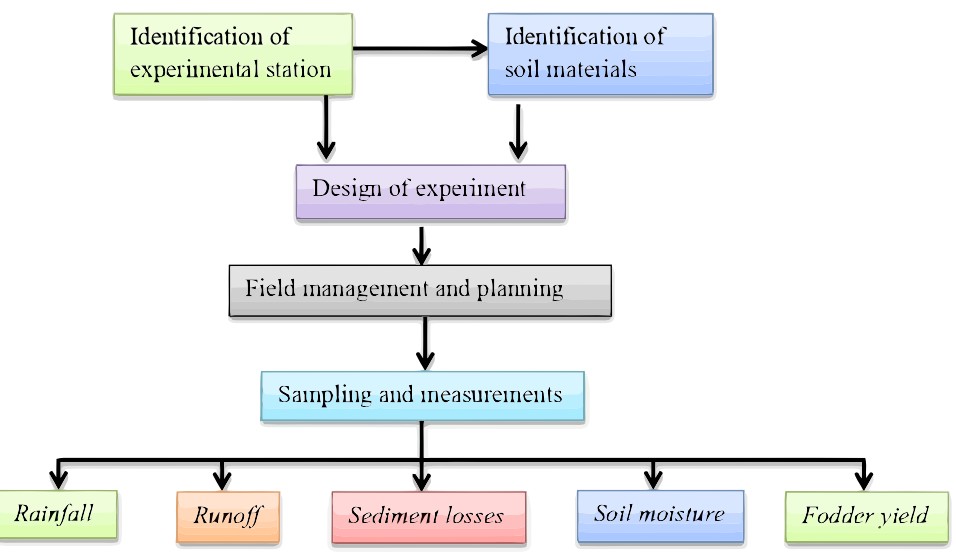

**Figure 1** **Schematic overview of the experimental program.**

tillage systems used, at a 7° slope. The biodegradable film (Ecoflex FS) used to mulch the ridges were 0.008 mm in thickness. Biochar applied in furrows was manufactured from maize straw at 400 °C through pyrolysis and thermal decomposition at Sanli New Energy Company in Henan, China, and applied at 30,000 kg ha$^{-1}$ in the fields (*Luo et al., 2017*).

## The experimental station

Field research was piloted at Anjiagou Catchment during alfalfa cultivation period from April–October (2020). The terrain of this area (latitude 35°34′N, longitude 104°39′E, and altitude 2,075 m a.s.l.) is mountainous with steep slopes (converted to grasslands after 'Grain-for-Green Policy' enacted in the 1990s). The experimental station is situated 2–3 km east of Dingxi city, Gansu Province, Northwest China (Fig. 2). The area is semi-arid with mean annual air temperature (7.2 °C) and monthly mean temperatures ranging from 1.1 °C in January to 19.1 °C in July. The soil type on the experimental station is calcic cambisol, according to American soil classifications (*Chen, Yang & Wei, 2013*). The soil chemical properties are outlined in Table 1. The farming practice in this area is monoculture with once a year crop harvesting due to low temperatures. The main crops grown in this area are proso millet (*Panicum miliaceum*), spring wheat (*Triticum aestivum*), potato (*Solanum tuberosum*), maize (*Zea mays*), and flax (*Linum usitatissimum*). The major fodder grass species are sainfoin (*Onobrychis viciifolia*) and alfalfa (*Medicago sativa*).

## Experimental design

In a completely randomized design, there were six plots (2 open / tied ridge with bio-degradable film cover ×2 biochar / no biochar + 2 flat planting (FP) as control) with three replications. Tied-ridging, open-ridging, and flat planting were the three tillage systems used, at a 7° slope. A ridge width, height, and furrow width of 45, 20, and 60 cm respectively, were used for open ridging and tied ridging. The ties in the tied ridging ranged from 10

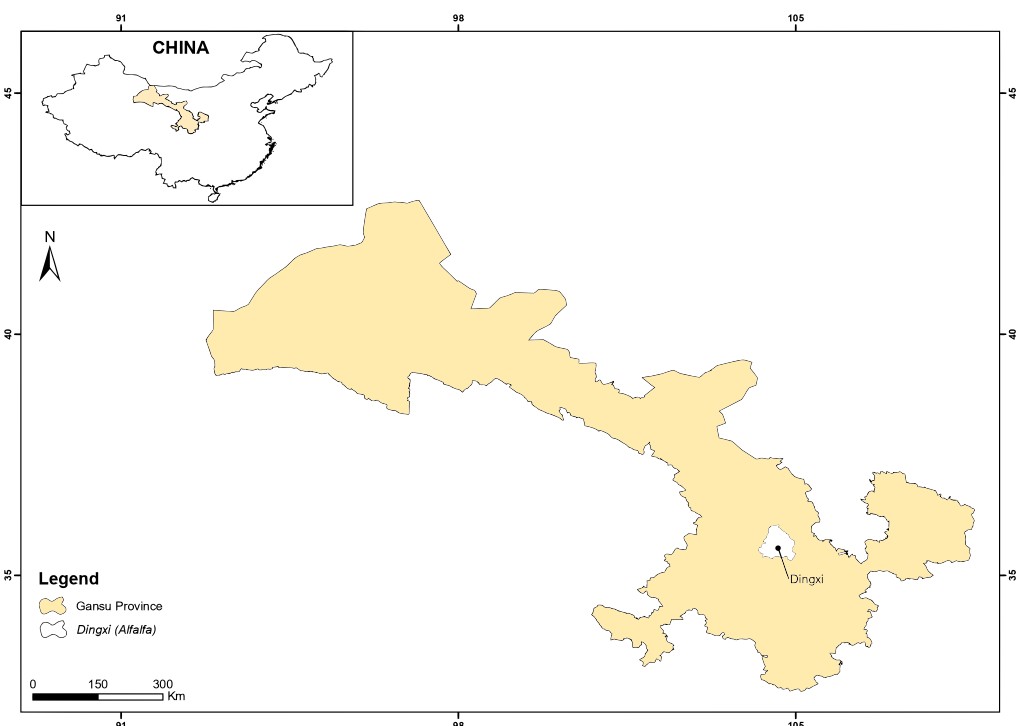

**Figure 2** **Experimental location of the study.** ArcGIS 10.6 software (ESRI, Redlands, CA, USA) was used to produce the map.

**Table 1** **Soil physical and chemical properties in the experimental field.**

| Depth (cm) | Bulk density (g cm⁻³) | Field capacity (%) | Total N (g kg⁻¹) | Total P (g kg⁻¹) | Total K (g kg⁻¹) | Organic matter (mg kg⁻¹) | Available N (mg kg⁻¹) | Olsen P (mg kg⁻¹) | Available K (mg kg⁻¹) | pH |
|---|---|---|---|---|---|---|---|---|---|---|
| 0–40 | 1.09 | 20.0 | 0.62 | 0.76 | 20.70 | 9.56 | 65.75 | 7.78 | 135 | 7.83 |
| 20–40 | 1.36 | 21.0 | 0.54 | 0.64 | 20.51 | 7.77 | 22.10 | 3.00 | 90 | 7.82 |

to 15 cm in height and 20 cm in width (Fig. 3). There was a 2.5 m distance between two non-staggered tied-ridges. The biodegradable film (Ecoflex FS) used to mulch the ridges were 0.008 mm in thickness. The bio-degradable film was mass-produced by BASF Co Ltd, Germany. Biochar was manufactured from maize straw at 400 °C through pyrolysis and thermal decomposition at the Sanli New Energy Company in Henan, China, and applied at 30,000 kg ha⁻¹ in the fields (*Luo et al., 2017*). Biochar had a specific surface area of 44 m² g⁻¹, a bulk density of 0.45 g cm⁻³, a pH (v/v 1:2.5 biochar: distilled water) of 7.5, a cation exchange capacity of 24.1 cmol kg⁻¹, a water holding capacity (24 h) of 288%, and Total C, and Total N content of 89 and 0.3%, respectively. In the exception of flat planting (control), experimental plots were 5.0 m wide and 10.05 m long, with nine ridges and 10 furrows (Fig. 3). Each plot was surrounded by a 15 cm high panel to accumulate runoff and sediment and to prevent runoff and sediment from adjacent plots. A gutter was built at the bottom of each plot to channel runoff and sediment into a pool with a volume of
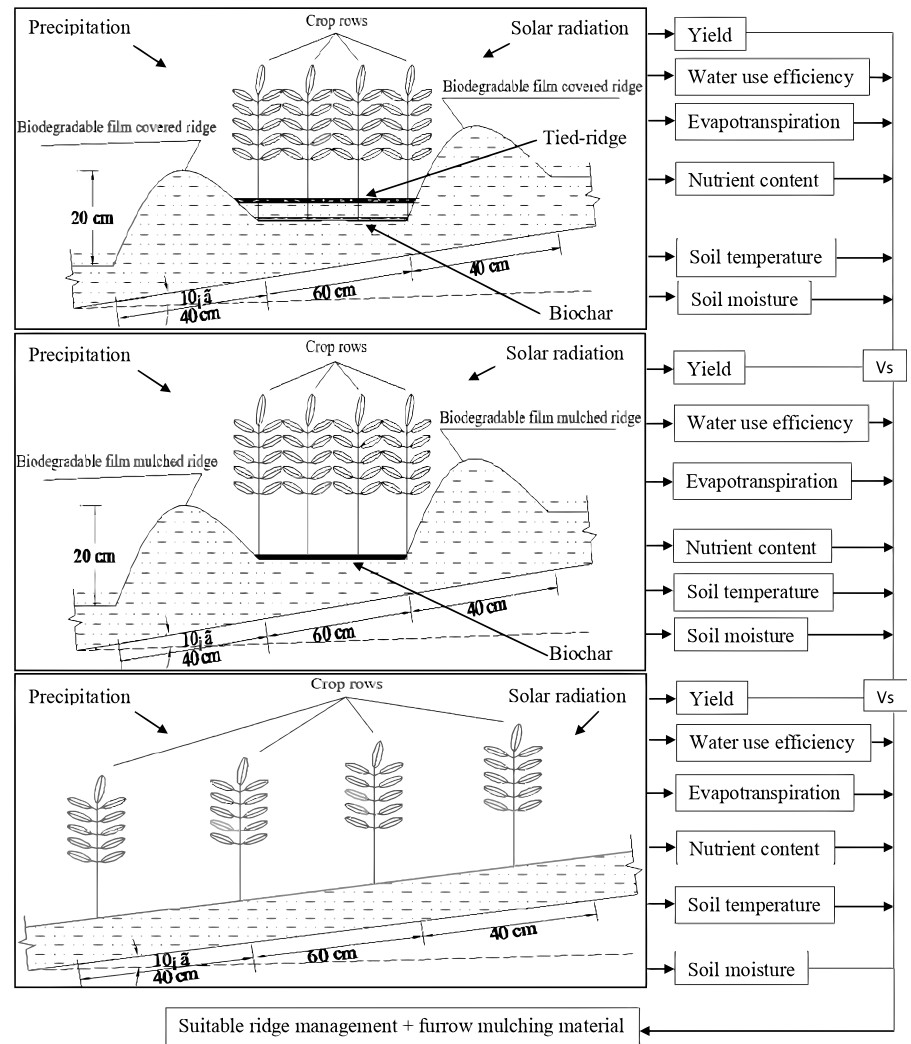

**Figure 3** Schematic diagram for alfalfa production in rainwater harvesting system with biochar amendment on sloping land.

2.25 m³ (1. 5 m wide × 1. 5 m long × 1.5 m deep). There was a 1.5 m space between two plots. Cement and bricks were used to build runoff and sediment collection pools to prevent infiltration.

## Field management

The fields were prepared when the soil was finally thawed on March 20, 2020, after clearing debris and litter. A 20 to 30 cm deep of high fertile soil was manually shoveled and piled up in accurate sizes and sloped with a tape measure and slope meter on April 2, 2020. Runoff, sediment collection pools, and boundaries were built on April 12, 2020. The furrows were used as planting zones after being ploughed, harrowed, and leveled. The bio-degradable film was laid on the ridges with edges buried 3 to 5 cm deep along ridge bases into the soil on April 12, 2020. Biochar was ground and screened through a 5-mm sieve before applied
to the field. On April 15, 2020, before seeding alfalfa, biochar was broadcasted by hand and promptly ploughed into a 0–20 cm soil depth. Localized alfalfa (No 3 Gannong) cultivar was cultivated on April 15, 2020, at 22.5 kg ha$^{-1}$. Four rows were sown in a 60 cm wide furrow, 2–3 cm deep with 20 cm spaces between 2 rows (Fig. 3). For tied ridging and open ridging, each experimental plot was 30 m$^2$, with 10 furrows (0.6 m width × 5 m length) and 40 alfalfa planted rows. Flat planting plots were 50.25 m$^2$ (5.0 width × 10.05 m length) with 66 alfalfa rows. Around 2 months after sowing, tied-ridges were manually built (June 14, 2020). Weeds were manually controlled with care to avoid breaking the ridges and no fertilizer or irrigation was carried out on experimental fields.

## Sampling and measurements
### Rainfall, runoff, and sediment losses
Data were collected as previously described in *Wang et al. (2018)*. Rainfall was measured on the experiment field with an automatic weather station (WSSTD1, England). Sediment debris in gutter was swept and collected with a broom and shovel into runoff and sediment pool after every main rainfall event. Runoff in the pool was calculated by multiplying inner basal area of the pool to runoff depth. Pool runoff was stirred with a shovel for 5 to 10 min for uniform suspension of soil particles in the water. Sampling was done immediately with three 1,000 mL measuring flasks, and samples were dried to clear and weighted to estimate sediment transport. Runoff and sediment pools were emptied and swept after sampling to provide space for the next runoff and sediment data sampling.

### Soil moisture
During alfalfa cultivation period, soil moisture was measured gravimetrically to a depth of 200 cm, with an increment of 20 cm at furrow bottom in each plot, at 10 days intervals, without considering soil moisture, before sowing or green-up and after cutting. Three random soil samples from top, middle and bottom (up-slope, middle-slope, and down-slope) of each plot were collected. The soil water content was determined in addition to other standard measurements on experimental plots, 24 hrs after every rainfall (>5 mm).

### Fodder yield
At the early flowering phase (between the first and 25% of flower) and senescence, alfalfa was manually harvested (cut) three times in all plots in 2020. After cutting, harvested alfalfa was immediately weighed, and 1 kg of the samples was dried in an oven at 105 °C for an hour and then at 75 °C for 72 h to measure alfalfa fodder yield. Alfalfa fodder yield was measured in 2 methods: (1) net fodder yield (NFY) in furrows (excludes tied-ridged areas); (2) actual fodder yield (AFY) in land areas of ridges and furrows (includes tied-ridge areas).
## Calculations of runoff and sediment parameters

Runoff, sediment yield, and runoff efficiency were calculated using these formulae.

$$V_{runoff} = A_{pool} \times D_{pool} \qquad (1)$$

$$W_{sediment} = V_{runoff} \times (W_{sample\,sediment} / V_{sample}) \qquad (2)$$

$$D_{runoff} = V_{runoff} / A_{plot} \qquad (3)$$

$$W_{sediment\,per\,area} = W_{sediment} / A_{plot} \qquad (4)$$

$$RE = V_{runoff} / (P \times A_{plot}) \qquad (5)$$

where $V_{runoff}$ ($m^3$) is pool runoff, $A_{pool}$ (2.25 $m^2$) is pool inner basal area, $D_{pool}$ (m) is pool runoff depth, $W_{sediment}$ (g) is pool sediment weight, $V_{sample}$ (L) is collected sample, $W_{sample\,sediment}$ (g) is sample sediment weight, $D_{runoff}$ ($Lm^{-2}$) is runoff depth, $A_{plot}$ ($m^2$) is plot projection area, $W_{sediment}$ per area ($gm^{-2}$) is sediment per area weight, RE (%) is runoff efficiency, and P (mm) is precipitation. The total actual evapotranspiration (ET, mm) for alfalfa cultivation period and water use efficiency (WUE, kg ha$^{-1}$ mm$^{-1}$) of alfalfa were calculated using these formulae (*Li & Gong, 2002*):

$$ET = P + (W_1 - W_2) \qquad (6)$$

$$WUE = NFY / ET \qquad (7)$$

$$WUE = AFY / ET \qquad (8)$$

where P is precipitation (mm) during alfalfa cultivation period, NFY (kg ha$^{-1}$) is net fodder yield, and AFY (kg ha$^{-1}$) is actual fodder yield. The filtration and recharge from groundwater are negligible in this area (*Zhao et al., 2012*). Soil moisture ($W_1$ and $W_2$) was also estimated with equation:

$$W = \theta_i \times \rho d_i \times H \times 10 \qquad (9)$$

Where $\theta$ is soil water content (%), H is soil profile thickness (cm); $\rho d$ is soil bulk density (g cm$^{-3}$).

## Statistical analysis

An SPSS statistical software package (version 26.0, SPSS Inc., IL, Chicago, USA) was used to analyze all the data. Differences between treatments were analyzed using a one-way analysis of variance (ANOVA) followed by Tukey Pairwise comparison at 5% significance and a linear regression analysis. The research location was mapped by GIS software (ESRI® ArcMap™ 9.3), and figures plotted by SigmaPlot 14.0 (Systat Software Inc., San Jose, California, USA).

# RESULTS

## Rainfall

Annual rainfall was 512.5 mm, with 451.2 mm falling during the alfalfa cultivation season (April 1 to October 9, Fig. 4). From January to December, monthly rainfall was 7.5, 4.7,

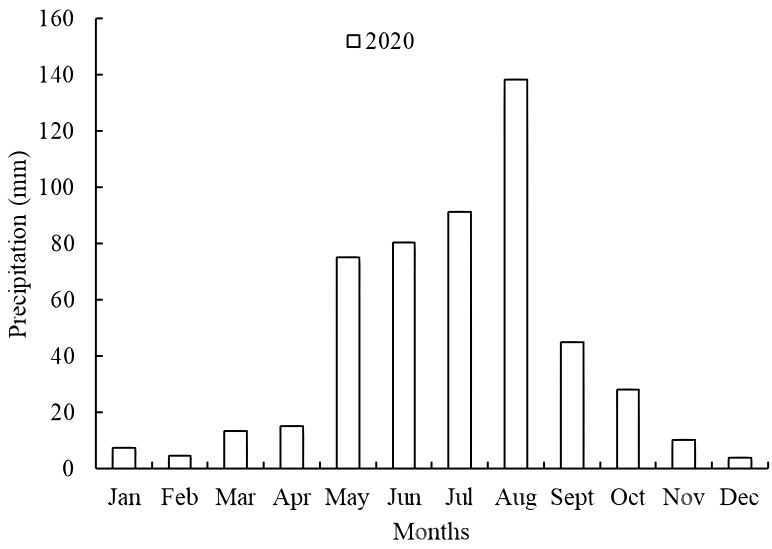

**Figure 4** **Mean monthly precipitations at the experiment station in 2020.**

13.4, 15, 75, 80.5, 91.2, 138.2, 44.8, 28.2, 10.2, and 3.8 mm, respectively. In the experimental year, rainfall from April to October accounted for 88 percent of total annual precipitation.

## Runoff, runoff efficiency, and sediment losses

Runoff in flat planting (30%), open ridging (45%), and tied ridging (52%) were decreased with biochar as sediment yield was decreased in flat planting (33%), open ridging (43%), and tied ridging (44%) (Fig. 5). Compared to flat planting, mean runoff was reduced in open ridging (38%) and tied ridging (55%) with biochar, and decreased in open ridging (20%) and tied ridging (33%) with no-biochar. Again, when compared to flat planting, sediment yield was considerably lower in open ridging (70%) and tied ridging (85%) with biochar, comparable to a drop in sediment production in open ridging (65%) and tied ridging (82%) with no-biochar. Runoff efficiency was decreased in open ridging (35%) and tied ridging (52%) with biochar amendment whereas runoff efficiency was decreased in open ridging (19%) and tied ridging (35%) with no-biochar compared to flat planting. Mean runoff efficiency was decreased in flat planting (31%), open ridging (45%), and tied ridging (50%) with biochar amendments compared to no-biochar. Thus, decrease in runoff and sediment in open and tied ridging rainwater harvesting methods may be attributed to decrease in runoff efficiency, as demonstrated by this experiment.

## Soil temperature

Mean soil temperatures on ridges and in furrows increased from April to July and then decreased until October during alfalfa growing season with biochar or no-biochar (Fig. 6). Mean soil temperatures on ridges and furrows in biochar plots were higher than in no-biochar plots. Concurrently, mean soil temperatures in biochar and no-biochar plots ranged from 14 °C to 26 °C. In furrows of biochar and no-biochar plots, mean soil temperatures varied from 12 °C to 23 °C and 12 °C to 24 °C, respectively. During alfalfa

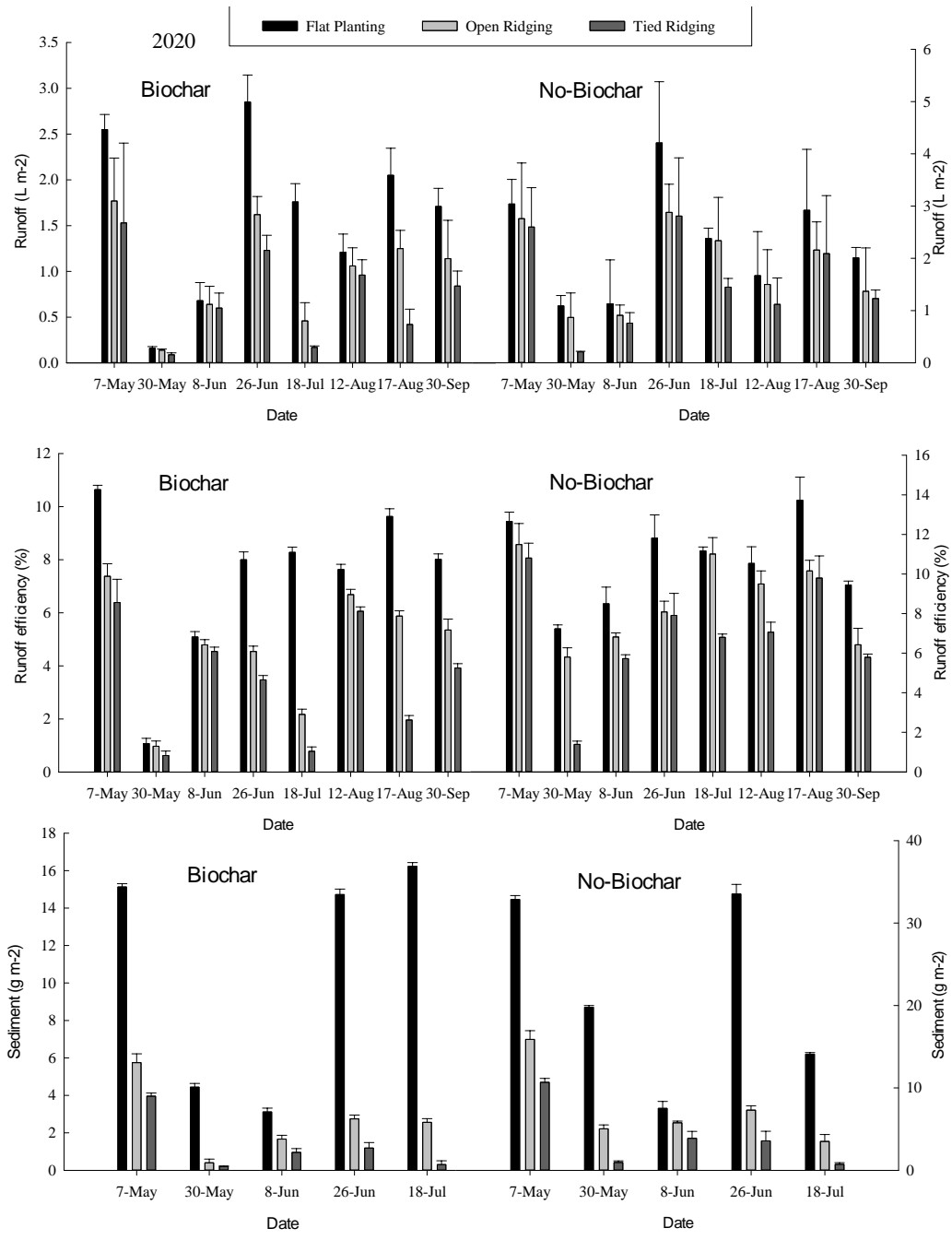

**Figure 5 Runoff, runoff efficiency and sediment in different treatments.** The means (columns) labeled with the same letters within each category are not significantly different at the 5% level (Tukey's-b test ANOVA).

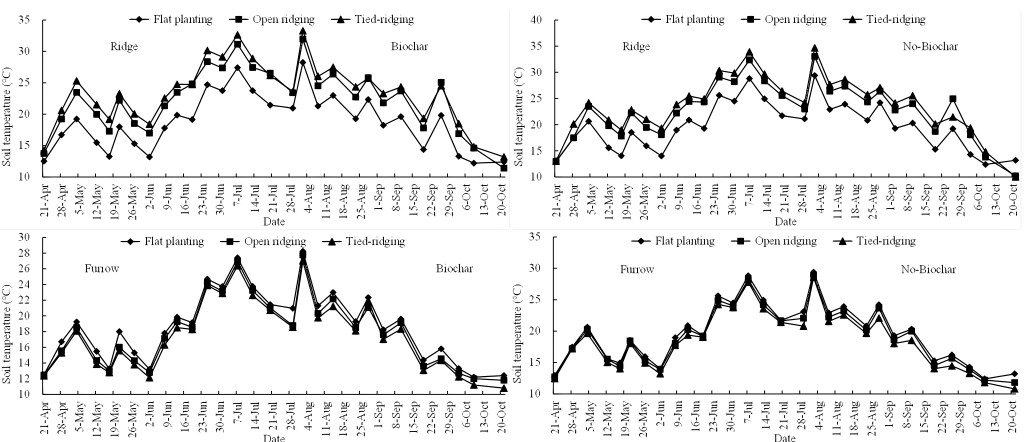

**Figure 6    Soil temperatures in furrows and on ridge tops in 0–25 cm soil depth in various treatments.**

cultivation period with biochar or no-biochar, soil temperature on ridges of tied ridging (TR) was significantly higher than that on open ridging (OR), which was significantly higher than that on flat planting (FP). Again, during alfalfa cultivation cycle with biochar, soil temperatures on ridges were significantly ($p = 0.00$; $F = 0.05$; R square = 0.006; Adjusted R Square = $-0.105$) increased in TR (24%) and OR (18%) relative to FP, and corresponding soil temperatures on ridges were increased in TR (20%) and OR (14%) in no-biochar plots. Meanwhile, during alfalfa cultivation period, soil temperature in furrows in biochar and no-biochar plots were in the order FP > OR > TR. Soil temperature was lower in TR (7%) and OR (4%) in biochar amended plots compared to FP, whereas in no-biochar plots, soil temperature was lower in TR (6%) and OR (4%). However, temperature differences in furrows with biochar amendment and in no-biochar plots were discovered to be non-significant ($p = 0.43$; $F = 0.86$; R square = 0.088; Adjusted R Square = $-0.014$) for TR, OR, and FP. According to findings from this study, decrease in soil temperature in tied ridging with biochar can be attributed to reduction in runoff and sediment.

## Soil water storage

Monthly soil water storage increased in tied ridging compared to open ridging, which was also higher than flat planting in biochar plots from April to June (Fig. 7). However, mean soil water storage increased in open ridging compared to tied ridging, which was higher than flat planting from July to October. The mean soil water storage was significantly higher in OR ($p = 0.00$) and TR ($p = 0.01$), as compared to FP in biochar amended plots ($F = 14.76$; R square = 0.48; Adjusted R Square = 0.39). In no-biochar plots ($F = 10.97$; R square = 0.65; Adjusted R Square = 0.59), mean soil water storage was significantly higher in OR ($p = 0.01$) and TR ($p = 0.00$), as compared to FP. During alfalfa cultivation period, mean soil water storage in middle-slope was higher than in down-slope, which was higher than in up-slope. The mean soil water storage for flat planting, open ridging,

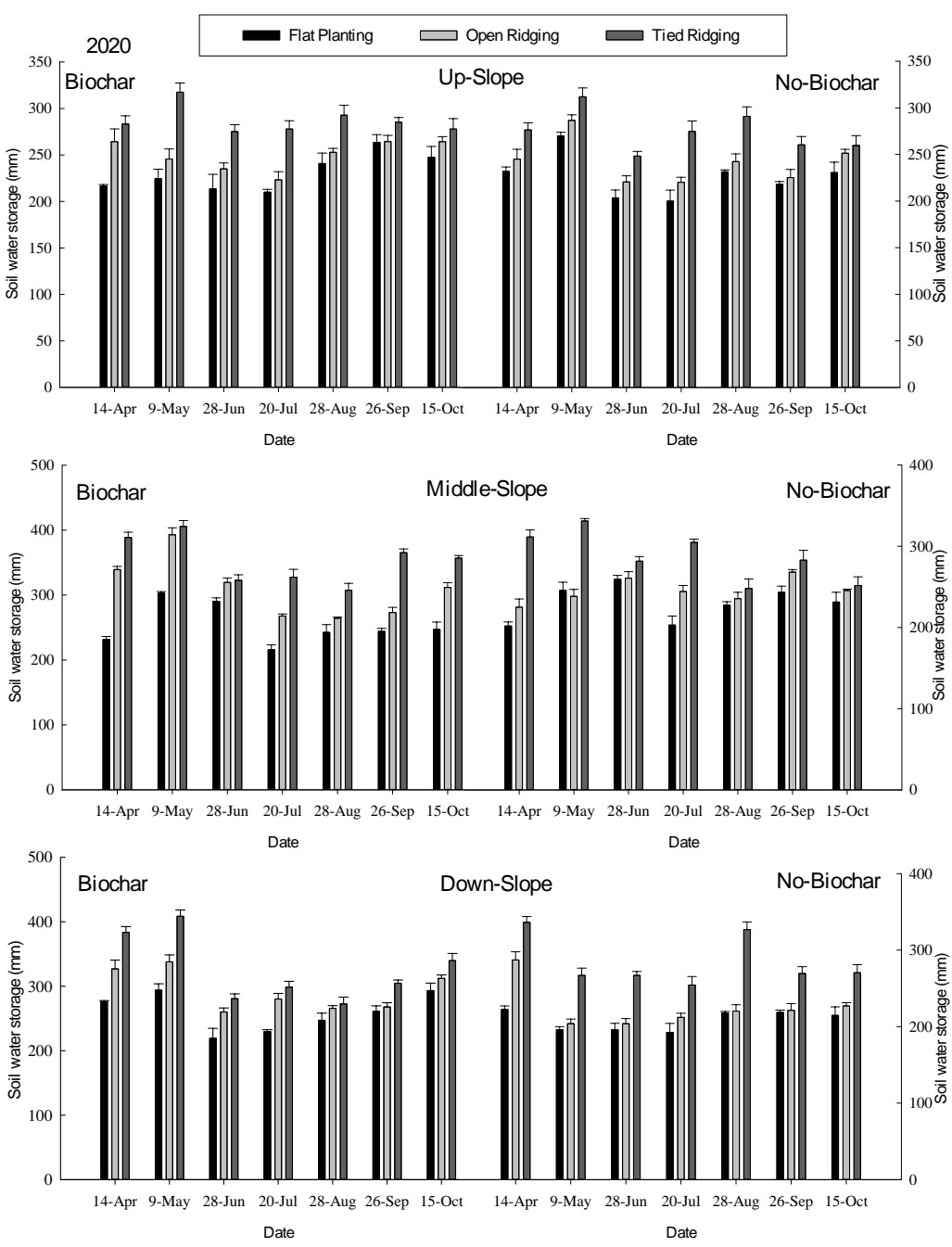

**Figure 7** **Soil water storage in furrows in 0–200 cm soil depth in various treatments.** The means (columns) labeled with the same letters within each group are not significantly different at the 5% level (Tukey's-b test ANOVA).

and tied ridging with biochar amendments was 243, 302, and 292 mm, respectively, while mean soil water storage for FP, OR, and TR with no-biochar was 232, 295, and 232 mm. In comparison to no-biochar, biochar amendments resulted in significantly ($p = 0.0$) higher mean soil water storage for FP, OR, and TR. This finding implies that biochar amendments

have a positive effect on soil water storage in open ridging compared to tied ridging. This could be attributed to lower runoff, sediment losses, and soil temperatures in the treatment fields.

## Fodder yield and water use efficiency

With no-biochar, NFY of first cut in TR was higher compared to OR, while OR was also higher compared to FP (Table 2). However, for second and third cuts, NFY in OR was higher than TR, which was also higher than FP in no-biochar plots. Cumulative annual NFY of the treatments were in the order TR > OR > FP in no-biochar plots. In biochar plots, NFY was significantly ($p = 0.00$; $F = 98.767$; R square = 0.971; Adjusted R Square = 0.961) higher among treatments in similar order as in no-biochar plots for first and second cuts. For third cut, NFY was in the order OR > TR > FP in biochar plots. Cumulative annual NFY increased in the order TR > OR > FP. The mean NFY was significantly higher ($p = 0.04$) in biochar amended plots than in no-biochar plots for all cuts. Consequently, TR had a significant effect on NFY in both biochar and no-biochar plots which could be due to increases in soil water storage in tied ridging.

Actual fodder yield was higher in TR which was higher than in OR which was higher than in FP for first cut with biochar amendments (Table 2). AFY was in the order OR > TR > FP in biochar plots for second and third cuts. Cumulative annual AFY with biochar amendments was significantly ($p = 0.02$) higher in TR than in OR, which was in turn significantly ($p < 0.0001$) higher than in FP. Furthermore, in no-biochar plots, AFY for first and third cuts was in the order OR > TR > FP. Subsequently, AFY was higher in all treatments with OR recording highest AFY whiles FP recorded lowest for second cut in no-biochar plots. Twelve-monthly AFY in no-biochar plots for all treatments was in the order OR > TR > FP. The mean actual fodder yield was significantly ($p = 0.00$; $F = 937.6$; R square = 0.99; Adjusted R Square = 0.996) higher in biochar amended plots than in no-biochar plots for all cuts. As a result, tied ridging had a significant effect on actual fodder yield in biochar plots, while open ridging had a significant effect on actual fodder yield in no-biochar plots. As demonstrated by this research, this can be attributed to a reduction in runoff and sediment losses, which lead to an increase in soil water storage in treatment fields.

Water use efficiency (WUE) was highly significant ($p < 0.0001$; $F = 1.460$; R square = 0.378; Adjusted R Square = 0.119) in tied ridging, compared to OR and FP in biochar and no-biochar plots (Table 2). Additionally, mean WUE was highly significant ($p = 0.01$; $F = 5.08$; R square = 0.62; Adjusted R Square = 0.505) in biochar plots than in no-biochar plots. Open ridging with biochar amendments increased net fodder yield (7.5%) compared to open ridging in no-biochar plots, while tied ridging with biochar amendments increased net fodder yield (8.5%) when compared to tied ridging in no-biochar plots. Open ridging with biochar amendments significantly ($p < 0.0001$; $F = 1187.047$; R square = 0.998; Adjusted R Square = 0.997) increased actual fodder yield (9.3%) compared to open ridging in no-biochar plots, while tied ridging with biochar amendments significantly ($p = 0.0001$; $F = 1187.047$; R square = 0.998; Adjusted R Square = 0.997) increased actual fodder yield (15.7%) compared to tied ridging in no-biochar plots. Biochar plots had a

Mak-Mensah et al. (2021), *PeerJ*, DOI 10.7717/peerj.11889

**Table 2   Alfalfa forage yield and water use efficiency (WUE) in tied-ridge-furrow rainwater harvesting with biochar amendment.**

| Biochar amendment patterns | Tillage practices | Fodder yield (kg ha$^{-1}$) | | | | | | | | WUE (kg ha$^{-1}$ m$^{-1}$) |
|---|---|---|---|---|---|---|---|---|---|---|
| | | First cut | | Second cut | | Third cut | | Annual total | | |
| | | NFY | AFY | NFY | AFY | NFY | AFY | NFY | AFY | |
| | | | | | 2020 | | | | | |
| | FP | 1728c | 1728c | 1027c | 1027c | 487c | 487c | 3242c | 3242c | 15.23c |
| Biochar | OR | 5726b | 2679b | 2527b | 1298a | 1198a | 597a | 9451b | 4574b | 28.49b |
| | TR | 5928a | 2826a | 2648a | 1283a | 1036b | 572b | 9612a | 4681a | 30.87a |
| | FP | 1628c | 1628c | 972c | 972c | 418c | 418c | 3018c | 3018c | 13.45c |
| No-Biochar | OR | 5289b | 2372a | 2486a | 1185a | 1013a | 627a | 8788b | 4184a | 22.49b |
| | TR | 5387a | 2267b | 2481a | 1190a | 987b | 589b | 8855a | 4046b | 24.73a |
| Mean | Biochar | 4461 | 2411 | 2067 | 1203 | 907 | 552 | 7435 | 4166 | 24.86 |
| | No-Biochar | 4101 | 2089 | 1980 | 1116 | 806 | 545 | 6887 | 3749 | 22.89 |

**Notes.**

[a] NFY (Net fodder yield) was forage yield based on furrow areas (exclude ridge and tied-ridge areas).

[b] AFY (actual fodder yield) was forage yield based on land areas of ridges (include ridge and tied-ridge) and furrows.

[c] FP, OR and TR were flat planting, open ridging and tied-ridging, respectively.

[d] Means within a column followed by the same letters are not significantly different at the 5% level (Tukey's-b test ANOVA).

higher average annual mean net fodder yield (8%) and actual fodder yield (11%) than in no-biochar plots. The increase in WUE with biochar amendments in tied ridging may be connected to the increase in yield and decrease in runoff and sediment, as demonstrated by this experiment.

## DISCUSSION

Poor and erratic rainfall in semi-arid areas is a challenge to rain-fed agriculture, where farmers may experience crop damage (*Graef & Haigis, 2001*). From this experiment, we found a decrease in runoff in flat planting (FP), open ridging (OR), and tied ridging (TR) by 30%, 45%, and 52%, respectively, and corresponding sediment yield decrease of 33%, 43%, and 44%, respectively, with biochar amendments (Please, refer to Fig. 5). When compared to no-biochar plots, mean runoff efficiency in flat planting, open ridging, and tied ridging was decreased by 31%, 45%, and 50%, respectively, with biochar amendments (Please, refer to Fig. 5). These results are in line with those of *Araya & Stroosnijder (2010)* who found runoff in tied ridges in a wheat field was significantly lower than runoff in a flat field. *Nuti et al. (2009)* elucidated that decreased runoff in tied ridges results in water storage in soil profiles which leads to improved crop development with higher crop yields. Furthermore, *Patil & Sheelavantar (2004)* reported decreased runoff with compartmental bunding and ridges and furrows relative to flat planting. Concurrently, tied-ridges decreased runoff by 51 and 58%, in Machanga, Kenya, during short and long rainy seasons, respectively (*Okeyo et al., 2014*). In a similar research in Upper Volta, tied ridges resulted in 0.9% runoff relative to 6.3% with open ridges and 12.2% in flat planting (*Gerbu, 2015*). *Xia et al. (2014)* came to a similar conclusion, finding a substantial reduction in runoff, phosphorus, and nitrogen losses. According to *Woldegiorgis (2017)*, effectiveness of tied ridging to decrease soil erosion was predominantly connected to decrease in runoff in tied ridge fields.

From this research, in biochar and no-biochar fields, we found soil temperatures on ridges of TR were significantly higher than on OR, which was significantly higher than on FP during alfalfa cultivation period. Conversely, soil temperature in furrows during alfalfa cultivation period in biochar and no-biochar fields were significantly higher in treatments in the order FP > OR > TR (Please, refer to Fig. 6). This finding is in agreement with *Genesio et al. (2012)*, who discovered dark-colored biochar increased soil temperatures compared to no-biochar. This may be attested to decrease in runoff and sediment with biochar amendment in this experiment.

Implementation of soil moisture preservation practices such as tied ridges with mulching has presented improved soil moisture retention in different environments (*Ndlangamandla, Ndlela & Manyatsi, 2016*). In comparison to no-biochar, mean soil water storage was found to be significantly higher in FP, OR, and TR with biochar (Please, refer to Fig. 7). This demonstrates the positive influence of biochar amendments in tied ridging on soil water storage. This result is in line with *Ndlangamandla, Ndlela & Manyatsi (2016)*, who found that soil moisture in the tied ridges with mulch was retained longer than in un-mulched ridges. These outcomes are also in agreement with *Mupangwa, Love & Twomlow (2006)* and *Nyamadzawo et al. (2013)*, who revealed tied ridges, are effective at trapping and

concentrating moisture in the root zone of plants. *Adimassu et al. (2014)* and *Al-Seekh & Mohammad (2009)* also reported lower runoff and higher soil moisture content. In similar research by *McHugh et al. (2007)*, open ridges performed second best with higher seasonal soil moisture than subsoiling, no-till, and traditional tillage. Again, *Araya & Stroosnijder (2010)* reported tied ridging with mulching can increase soil water in the root zone of plants by 13% and crop grain yield (barley) by 44% during below-average rainfall years. Conversely, tied ridging with soil bund and tied ridging with mulch increased soil water storage by 5.37% and 6.20%, respectively, compared to flat planting (*Adeboye et al., 2017*).

In this study, annual cumulative net fodder yield (NFY) was significantly higher in TR, which was significantly higher compared to OR, which too was significantly higher compared to FP (Please, refer to Table 2). This is in line with *Zelelew, Ayimute & Melesse (2018)*, whose findings indicated that plots treated with tied ridge had the highest grain yield (2,302 kg ha$^{-1}$) and biomass (7,647 kg ha$^{-1}$). Grain yield for tied ridges with mulch was substantially different ($p < 0.05$) from bare tied ridges and flat planting, according to *Ndlangamandla, Ndlela & Manyatsi (2016)*. The significant difference among these treatments is agreed to be as a result of moisture retention which was attained as an effect of mulching (*Ndlangamandla, Ndlela & Manyatsi, 2016*). In another study, pearl millet yield was significantly increased in tied ridging than in flat planting (*Silungwe et al., 2019*). Tied ridging has been successful in other semi-arid areas for cereals like sorghum (*Sorghum bicolor*) (*Mesfin et al., 2009*; *Bayu, Rethman & Hammes, 2012*). The yield of crops (sorghum, maize, wheat, and mung bean) grown with tied ridging significantly increased (50 to 100%) as compared to flat planting in semi-arid areas (*Zelelew, Ayimute & Melesse, 2018*). Correspondingly, relative to flat planting, furrow planting in open-end tied ridges resulted in a 28.86% increase in stover yield (*Belachew & Abera, 2014*). The grain yield harvested in tied ridging (3.6 t ha$^{-1}$) was higher (12.5%) compared to flat planting (3.2 t ha$^{-1}$) (*Yoseph, 2014*). Furthermore, maize biomass yield (11,019 kg ha$^{-1}$) in closed-end tied ridging was highest with a 54.9% increase compared to flat planting (*Belachew & Abera, 2014*). In addition, *Sumeriya, Singh & Kaushik (2014)* revealed an increase in sorghum grain yield ranging from 67 to 73% and soil water (40%) in tied ridging compared to flat planting. As a consequence, depending on rainfall and slope gradient, tied ridging has been shown to increase yields (*Motsi, Chuma & Mukamuri, 2004*; *McHugh et al., 2007*).

With biochar amendments, tied ridging had a significant effect on actual fodder yield (AFY), while OR had a significant effect on AFY with no biochar. Biochar improved the annual cumulative mean of NFY (8%) and AFY (11%) as compared to no-biochar in this study (Please, refer to Table 2). These outcomes are in line with *Mak-Mensah et al. (2021)*, who reported combined application of biodegradable film with biochar in the Loess Plateau of China increased yield by 22.86% compared with FP. This was corroborated by *Liu et al. (2014)* who achieved a higher yield of sweet potato (53.77%; $p < 0.05$), with biochar amendment than with no biochar treatment (control). In addition, *Liang et al. (2014)* obtained a 10% increase in grain yield in winter wheat and summer maize with biochar application compared to controls (no biochar). Furthermore, *Xiao et al. (2016)* found that 20 and 30 t ha$^{-1}$ biochar amendment improved wheat yields by 9 and 13% in 2012 and 11 and 14% in 2013, respectively, compared to no biochar treatments. In

comparing biodegradable film mulched ridge-furrow with 20 t ha$^{-1}$ biochar application to biodegradable film mulched ridge-furrow without biochar treatments, wheat grain yield increased by 6 and 9% in 2012 and 2013 (*Xiao et al., 2016b*). In addition, a meta-analysis by *Jeffery et al. (2011)* found that biochar-treated soils enhanced crop productivity by 10% on average when compared to plots without mulching. Under co-application of biodegradable film mulched ridge-furrow with biochar treatment, the residual impact of biochar on soil fertility accounted for the majority of improvement in crop production (*Rehman & Razzaq, 2017*).

Improving water use efficiency in semi-arid regions can be attained either by increasing the volume of water accessible to plants for transpiration and/or by increasing efficacy with which transpired water yields more plant biomass (*Wallace, 2000*). Water use efficiency was in the order TR > OR > FP with no-biochar or biochar amendments (Please, refer to Table 2). The mean WUE was significantly higher in biochar plots than in non-biochar plots in this present research. These outcomes are consistent with *Ndlangamandla, Ndlela & Manyatsi (2016)* who reported increased soil moisture and crop yield with mulching in tied-ridging in Swaziland's semiarid areas. This may be an ideal agronomic practice for smallholder farmers to increase yield in crop production. The practice could also be used as a soil and water conservation strategy in rain-fed agriculture, especially in climate-changing areas to reduce drought impact while decreasing runoff and erosion (*Mak-Mensah et al., 2021*).

## CONCLUSIONS

The tied ridge with biochar amendments in alfalfa cultivation has been shown to reduce runoff and significantly improve rainfall infiltration into the soil. Field investigation revealed biochar amendments reduced runoff in flat planting, open ridging, and tied ridging, resulting in a decrease in sediment yield. Mean runoff efficiency was decreased in flat planting, open ridging, and tied ridging, with biochar amendments compared to no-biochar. During the alfalfa cultivation period with biochar or no-biochar, soil temperature on tied ridging ridges was significantly higher than that on open ridging, which was significantly higher than flat planting. In comparison to no-biochar, mean soil water storage for flat planting, open ridging, and tied ridging with biochar was significantly higher. This signifies the viability of biochar amendment in improving soil water storage in open ridging. Biochar increased annual cumulative net fodder yield and actual fodder yield means compared to no-biochar. Conversely, mean water use efficiency with biochar amendment was significantly higher than in no-biochar. Thus, when crop production is threatened by soil erosion and drought, tied ridging with biochar is beneficial to crop growth in rain-fed agriculture.

The study's main constraints were labor costs for creating ties, filming, and applying biochar, all of which are time-consuming tasks that need manual work. Other constraints discovered during this investigation include limited access to farm inputs, and the high cost of biochar and biodegradable film, low soil fertility and lack of fertilizer application. Thus, there is a great need to examine nutrient loss reductions of alfalfa cultivation under

a wide range of growing conditions and locations. Further research is needed on better plant water uptake cultivars or species, strategies for minimizing unproductive water losses, and consistent rodents and weeds control. In addition, the economic viability of alfalfa cultivation in tied-ridges with biochar amendments needs to be evaluated in response to drastic increases in input costs. Although, smallholder farmers in semi-arid areas could be trained in the use of this water-saving technique to reduce runoff, soil erosion, sediment losses, and improve food security, to overcome cultural and sociological reluctance in both rural and urban communities to deploy and accept this system, new ways for disseminating knowledge about tied-ridges with biochar amendments are needed.

### Funding
This research was funded by the National Natural Science Foundation of China (42061050 and 41661059). The funders had no role in study design, data collection and analysis, decision to publish, or preparation of the manuscript.

### Grant Disclosures
The following grant information was disclosed by the authors:
National Natural Science Foundation of China: 42061050, 41661059.

### Competing Interests
The authors declare there are no competing interests.

### Author Contributions
- Erastus Mak-Mensah conceived and designed the experiments, performed the experiments, analyzed the data, prepared figures and/or tables, authored or reviewed drafts of the paper, and approved the final draft.
- Faisal Eudes Sam analyzed the data, authored or reviewed drafts of the paper, and approved the final draft.
- Itoba Ongagna Ipaka Safnat Kaito analyzed the data, prepared figures and/or tables, and approved the final draft.
- Wucheng Zhao, Dengkui Zhang, Xujiao Zhou, Xiaoyun Wang and Xiaole Zhao performed the experiments, prepared figures and/or tables, and approved the final draft.
- Qi Wang conceived and designed the experiments, prepared figures and/or tables, and approved the final draft.

### Data Availability
   The raw measurements are available in the Supplementary File.

### Supplemental Information
Supplemental information for this article can be found online at http://dx.doi.org/10.7717/peerj.11889#supplemental-information.

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
