# Peer review of "Influence of tied-ridge with biochar amendment on runoff, sediment losses, and alfalfa yield in northwestern China"

_PeerJ, doi:10.7717/peerj.11889_

## Round 0.1 · original submission · Major Revisions

Authors, please kindly find reviewers' comments, which the editor considers very useful to improve the quality of your work. Kindly attend to them very diligently. Look forward to your revised manuscript. Thank you.

Reviewer 1 ·

Basic reporting

no comment

Experimental design

no comment

Validity of the findings

no comment

Additional comments

The study investigated how tied-ridge with biochar amendment improved soil characteristics and alfalfa yield in a specific area I china. The findings of the research were positive and significant. The quality of the work is of an acceptable standard. Some comments to improve the work are given below
1. Line 60, kindly include the botanical name of alfalfa at first mention.
2. Line 133-135, this need not appear in the introduction.
3. Line 142-145, to help readers appreciate the significance of the research more, it would be beneficial to include a map of the study area.
4. Line 162, if available, please give the product name of this biodegradable film. This could assist in cases where the experiments need to be replicated.
5. No details on biochar source, properties, and application technique, and quantity were presented in the methods section. In fact, there was no mention of the word ‘biochar' in this section. That is quite odd.
6. Line 277, 296, 339, 353, 356, 373, kindly avoid the use of personal pronouns like ‘we’ and ‘our’.
7. The manuscript should be thoroughly spell-checked and grammar-checked.

Reviewer 2 ·

Basic reporting

The English language needs to be improved to ensure that an international audience can clearly understand your text. Some examples where the language could be improved include lines 97 and 372. I suggest you have a proficient English colleague and familiar with the subject matter, reviews your manuscript, or use grammar checking tools.
One example can be described as lines 222 and 238, in which short sentences can be managed in tables.

Figure 2 and Figure 5 do not have high resolution to identify the contents. It is recommended to revise the figures.

Experimental design

The experimental design is well organized, which demonstrated consistency with other authors' findings. It can also be of help if authors provide findings in disagreement from the literature if applicable.

Validity of the findings

The findings are valid, well compared with other results from the literature.

·

Basic reporting

No comment.

Experimental design

No comment

Validity of the findings

No comment

Additional comments

This is a well written, clear and concise manuscript. The overall conclusions of the study are not novel, but the methodology is proper and the quality and amount of results, as well as their analysis is adequate. The study is interestring and the manuscript is worth of publication.

1 - Introduction: The agronomic benefits from the biochar addition to soil are widely acknowledge. However, biochar addition to soil can also present possible detriments to soil health. Such possible detriments should be referred by the authors. For example, the long-term effects of biochar application on soil including its fate in different soil types and management practices still need to be explored, such as influencing soil hydrophobicity and increasing soil sorption capacity of trace contaminants and contamination of aquifers. See Solaiman, Z.M. and Anawar, H.M. 2015. Application of Biochars for Soil Constraints: Challenges and Solutions. Pedosphere 25(5), 631–638. https://doi.org/10.1016/S1002-0160(15)30044-8;

2 - A figure of the location of the study area should be added.

3 - L156: "(3 tillage systems x 2 mulching materials + flat planting (FP) as control)" This is not well explained. This gives 7 plots. What do you mean with mulching materials? Biochar and no biochar, or biochar and the plastic? Be clearer.

4 - Figure 1: Figure 1 is not clear. What do the up and down arrows (on the left of the terms soil moisture, fodder yield and so on) mean? Represent the preferential direction of runoff and the slope gradient in the plots. Identify the biochar, the plastic mulch as well as the areas with the ridges, furrows and the lines with the crops and so on. The figure should be explanatory by itself.

5 - Equation 4: Capital W in Wsediment. Equation 5: It should be a capital P for precipitation. L227: Define WUE. Equation 6: Capital RE for runoff efficiency. Equation 10: Rectify the â symbol.

6 - The biochar characteristics and properties are not mentioned in the methodology. Also, the authors always refer the biochar application as mulching. Mulching is the biochar and the biodegradable film. In the title the authors refer biochar amendment. However, in the methodology, it seems that the authors forgot the main purpose of the article: study the influence of biochar amendment.

7 - L257-258: "The annual precipitation averaged 385.3 mm over 46 years, as rainfalls from May to August were higher than in the 46-years monthly record." What is the purpose of this information? And why a figure (Figure 2) detailing this? Is this important for this study?

8 - The authors should first introduce a Figure and then analyse it.

9 - Consider only "Conclusions" instead of "Conclusions and recommendations". The main reason is that the recommendations part is almost non-existent. Remove the numerical values from the conclusions. The authors already presented a lot of numerical values and percentages in the results. For the conclusions, it is only important to know if one treatment is better than other and at what.

10 - Figures 3-5 have very poor quality.

---

## Round 0.2 · Minor Revisions

Reviewers considered the revised manuscript improved, one indicated acceptance, the other indicated minor revisions, and provided comments. Please, authors, in addition, the editor suggests you kindly address the following :

1) Abstract: The last sentence(s) in the background should succinctly include how the hypothesis of this work points to the specific objective.

2) Introduction: Provide more information about tied-ridges. Provide more information on the importance of biochar amendments, and its implications.

Also provide more information as to why understanding the tied-ridge with biochar amendment is important, and if it has been evidenced, why is it relevant to be studied again. Consider the context of this study when making this argument (This aspect should be merged somehow with last paragraph, to justify the relevance of this work).

3) Materials and Methods:
-Given the nature of this work, kindly start this section with subsection called 'Schematic overview of the experimental program'. It should have t3-4 sentences, which should provide a snapshot of the experimental study. Make sure to have flow diagram, which could follow: identification of experimental station>identification of soil materials> design of experiment> field management and planning> sampling and measurements (apply your discretion on how to break down this last one)
-Calculations of runoff and sediment parameters (not 'Calculations' only)
-Statistics, why did you use general linear model 279and univariate ANOVA ? How did you resolve mean differences?

4) Results are ok. In all the probability levels, please you must provide the exact p-values, and F-values, R-sq (adjusted) values, at every single place where significance is shown. You used SPSS, so the SPSS must have provided these values. Where p>0,05, please provide these statistics as well. Make sure you clearly refer to either Table or Figure as well, where applicable.

5) Discussion is ok. Kindly make sure to have (Please refer to Table ??) or (Please refer to Figure ??) where discussions involve results specific to either figure or table.

6) Conclusions appear ok. Authors, kindly brainstorm - what were the limitations of this work? Any limitations? What have you learned from this work? What should be the direction of future studies?

Please make the conclusions two paragraphs, where paragraph one ties up the work, with what led to this study, why this study is relevant, key results that fill the gap (some are already in the current conclusions), paragraph two will then have some additions of key aspects of the work not mentioned in paragraph one, then what were the limitations of this work? Any limitations? What have you learned from this work? What should be the direction of future studies?

Looking forward to your revised manuscript. Please, make sure to attend to all these, one by one. The editor will be looking out for every single point raised here.

Thank you very much.

Reviewer 1 ·

Basic reporting

OK

Experimental design

OK

Validity of the findings

OK

Additional comments

The paper is now acceptable for publication.

Reviewer 2 ·

Basic reporting

Please correct line 136 by removing "to" in "can to". And please double-check for punctuations such as commas, for example, in line 436, before "according to."

Experimental design

The experimental results support research questions by investigating the knowledge gap with the help of a literature review.

Validity of the findings

Conclusions based on the data, which are robust and statistically sound, are acceptable.

---

## Round 0.3 · accepted · Accept

Thank you authors for revising your work and addressing all concerns raised by the reviewers. The authors have benefited from the peer review process and improved the quality of their work. The revised manuscript can now be accepted for publication. This is very fine work. Thank you for finding PeerJ as your journal of choice, and looking forward to your future scholarly contributions. Congratulations and very best wishes.